# SARS-CoV-2-specific T cell memory is sustained in COVID-19 convalescent patients for 10 months with successful development of stem cell-like memory T cells

Jae Hyung Jung[1,6], Min-Seok Rha [1,2,6], Moa Sa [1,3], Hee Kyoung Choi[4], Ji Hoon Jeon[4], Hyeri Seok[4], Dae Won Park[4], Su-Hyung Park [1,3], Hye Won Jeong [5✉], Won Suk Choi [4✉] & Eui-Cheol Shin [1,3✉]

Memory T cells contribute to rapid viral clearance during re-infection, but the longevity and differentiation of SARS-CoV-2-specific memory T cells remain unclear. Here we conduct ex vivo assays to evaluate SARS-CoV-2-specific CD4[+] and CD8[+] T cell responses in COVID-19 convalescent patients up to 317 days post-symptom onset (DPSO), and find that memory T cell responses are maintained during the study period regardless of the severity of COVID-19. In particular, we observe sustained polyfunctionality and proliferation capacity of SARS-CoV-2-specific T cells. Among SARS-CoV-2-specific CD4[+] and CD8[+] T cells detected by activation-induced markers, the proportion of stem cell-like memory T ($T_{SCM}$) cells is increased, peaking at approximately 120 DPSO. Development of $T_{SCM}$ cells is confirmed by SARS-CoV-2-specific MHC-I multimer staining. Considering the self-renewal capacity and multipotency of $T_{SCM}$ cells, our data suggest that SARS-CoV-2-specific T cells are long-lasting after recovery from COVID-19, thus support the feasibility of effective vaccination programs as a measure for COVID-19 control.

[1] Graduate School of Medical Science and Engineering, Korea Advanced Institute of Science and Technology (KAIST), Daejeon, Republic of Korea. [2] Department of Otorhinolaryngology, Yonsei University College of Medicine, Seoul, Republic of Korea. [3] The Center for Epidemic Preparedness, KAIST, Daejeon, Republic of Korea. [4] Division of Infectious Diseases, Department of Internal Medicine, Korea University College of Medicine, Ansan Hospital, Ansan, Republic of Korea. [5] Department of Internal Medicine, Chungbuk National University College of Medicine, Cheongju, Republic of Korea. [6] These authors contributed equally: Jae Hyung Jung, Min-Seok Rha. ✉email: hwjeong@chungbuk.ac.kr; cmcws@korea.ac.kr; ecshin@kaist.ac.kr

Severe acute respiratory syndrome coronavirus 2 (SARS-CoV-2) infection causes coronavirus disease 2019 (COVID-19), an ongoing pandemic disease that threatens public health[1]. As of May 2, 2021, more than 150 million confirmed cases had been reported, and over 3 million deaths worldwide[2]. After SARS-CoV-2 infection, some patients, particularly elderly patients, develop severe COVID-19 that is associated with hyper-inflammatory responses[3,4]. Global efforts are underway to prevent the transmission of SARS-CoV-2 and to develop novel vaccines and therapeutic strategies. A thorough understanding of the immune responses against SARS-CoV-2 is urgently needed to control the COVID-19 pandemic.

Increasing evidence has demonstrated that SARS-CoV-2-specific memory T cell responses are elicited after recovery from COVID-19. A number of studies have reported SARS-CoV-2-specific memory T cell responses in the early convalescent phase of COVID-19[5–9]. SARS-CoV-2-specific CD4+ and CD8+ T cells have been detected in 100% and ~70% of convalescent individuals a short time after resolution[5]. Recently, memory T cells were shown to contribute to protection against SARS-CoV-2 re-challenge in a rhesus macaque model[10]. Considering that T cell responses to SARS-CoV-1 and Middle East respiratory syndrome coronavirus (MERS-CoV) are long-lasting, up to >17 years[6,11–13], SARS-CoV-2-specific memory T cells are expected to be maintained long-term and to contribute to rapid viral clearance during re-infection. A very recent study has examined SARS-CoV-2-specific T cell responses up to 8 months after infection using activation-induced marker (AIM) assays[14].

Following natural infection or vaccination, the generation of effective and persistent T cell memory is essential for long-term protective immunity to the virus. Among diverse memory T cell subsets, stem cell-like memory T ($T_{SCM}$) cells were recently reported to have the capacity for self-renewal and multipotency to repopulate the broad spectrum of memory and effector T cell subsets[15,16]. Thus, the successful generation of $T_{SCM}$ cells is required for long-term protective T cell immunity[16]. For example, long-lived memory T cells following vaccination with live-attenuated yellow fever virus (YFV) exhibit stem cell-like properties and mediate lifelong protection[17,18]. However, limited knowledge is available on the differentiation of SARS-CoV-2-specific memory T cells following recovery from COVID-19, particularly the generation of $T_{SCM}$ cells.

In the present study, we report SARS-CoV-2-specific CD4+ and CD8+ T cell responses in peripheral blood mononuclear cells (PBMCs) from individuals infected with SARS-CoV-2 over 10 months post-infection. Using diverse T cell assays, we show that SARS-CoV-2-specific memory T cell responses are maintained 10 months after the infection. In addition, we report the differentiation of SARS-CoV-2-specific memory T cells during the study period and the successful generation of $T_{SCM}$ cells. We also demonstrate the effector functions and the proliferation capacity of long-term memory CD4+ and CD8+ T cells. Our findings provide insights for understanding long-term SARS-CoV-2-specific T cell immunity.

## Results

**Study cohort.** We recruited 101 individuals with SARS-CoV-2 infection. The peak disease severity was evaluated according to the NIH severity of illness categories[19]: asymptomatic ($n = 7$), mild ($n = 46$), moderate ($n = 25$), severe ($n = 14$), and critical ($n = 9$). Whole blood samples were obtained longitudinally (2–4 time points) from 56 patients or at a single time point from 45 patients. Whole blood was collected 1–317 days post-symptom onset (DPSO). Finally, a total of 193 PBMC samples were analyzed. Among 193 PBMC samples, 37 samples were obtained in

the acute phase when viral RNA was still detected (1–33 DPSO), and 156 samples were obtained in the convalescent phase after the negative conversion of viral RNA (31–317 DPSO). In the current study, we defined the acute phase as 1–30 DPSO, and the convalescent phase as 31–317 DPSO. The demographic and clinical characteristics of enrolled patients are presented in Supplementary Table 1. The proportion of patients with asymptomatic, mild, moderate, severe, and critical diseases at T1 (31–99 DPSO), T2 (100–199 DPSO), and T3 (≥200 DPSO) is presented in Supplementary Table 2. Details of the blood samples used in each assay are summarized in Supplementary Data 1.

**SARS-CoV-2-specific T cell responses are sustained 10 months after the infection.** First, we performed direct ex vivo interferon-γ (IFN-γ) enzyme-linked immunospot (ELISpot) assays following stimulation of PBMCs with overlapping peptide (OLP) pools spanning the spike (S), membrane (M), and nucleocapsid (N) proteins of SARS-CoV-2. S-, M-, and N-specific IFN-γ spot numbers increased during the acute phase. Subsequently, there was a decreasing tendency until 60-120 DPSO, and the IFN-γ responses were maintained over 10 months (Fig. 1a). These kinetics were also observed when S-, M-, and N-specific IFN-γ spot numbers were summed (Fig. 1a). In each PBMC sample, all three OLP pools evenly contributed to the IFN-γ responses without antigen dominance (Fig. 1b). We divided convalescent samples into three groups based on the DPSO at sample collection: T1 (31–99 DPSO), T2 (100–199 DPSO), and T3 (≥200 DPSO). We found no significant difference in IFN-γ spot numbers among groups (Fig. 1c), and a maintenance of an even contribution of antigens for the IFN-γ response among groups (Supplementary Fig. 1). Next, we focused on longitudinally tracked samples from 39 individuals in the convalescent phase. S-, M-, and N-specific and summed IFN-γ spot numbers were stable during the convalescent phase (Fig. 1d). When we compared paired samples from the same patient at two-time points (t1, 31-100 DPSO; t2, ≥ 200 DPSO), we found no significant difference in IFN-γ spot numbers (Fig. 1e).

To investigate the SARS-CoV-2-specific CD4+ and CD8+ T cell responses separately, we evaluated SARS-CoV-2-specific T cell responses by AIM assays[5,20,21] following stimulation of PBMCs with OLP pools of S, M, and N. CD137+OX40+ cells were considered SARS-CoV-2-specific cells among CD4+ T cells, and CD137+CD69+ cells were considered SARS-CoV-2-specific cells among CD8+ T cells[5] (Fig. 2a, b). We confirmed a strong positive correlation between the frequency of CD137+OX40+ cells and the frequencies of alternative AIM+ cells (OX40+CD154+ or CD137+CD154+ cells) among CD4+ T cells (Supplementary Fig. 2). The frequency of S-, M-, and N-specific CD137+OX40+ cells among CD4+ T cells increased during the acute phase, decreased until 60 DPSO, and then was maintained over 10 months (Fig. 2c). When the frequency of S-, M-, and N-specific CD137+OX40+ cells was summed, similar kinetics were observed (Fig. 2c). The frequency of CD137+CD69+ cells among CD8+ T cells was relatively low compared to the frequency of CD137+OX40+ cells among CD4+ T cells, but exhibited similar kinetics (Fig. 2c). We also observed a positive correlation between the frequency of CD137+OX40+ cells among CD4+ T cells and the frequency of CD137+CD69+ cells among CD8+ T cells (Supplementary Fig. 3).

We also measured the level of SARS-CoV-2 S receptor-binding domain (RBD)-specific IgG antibodies and SARS-CoV-2 neutralizing activity in plasma samples from COVID-19 convalescent patients. We found that the level of RBD-specific antibodies decreased over time, whereas the level of neutralizing activity was maintained (Supplementary Fig. 4a). In statistical analysis of time points T1 (31–99 DPSO), T2 (100–199 DPSO), and T3 (≥200

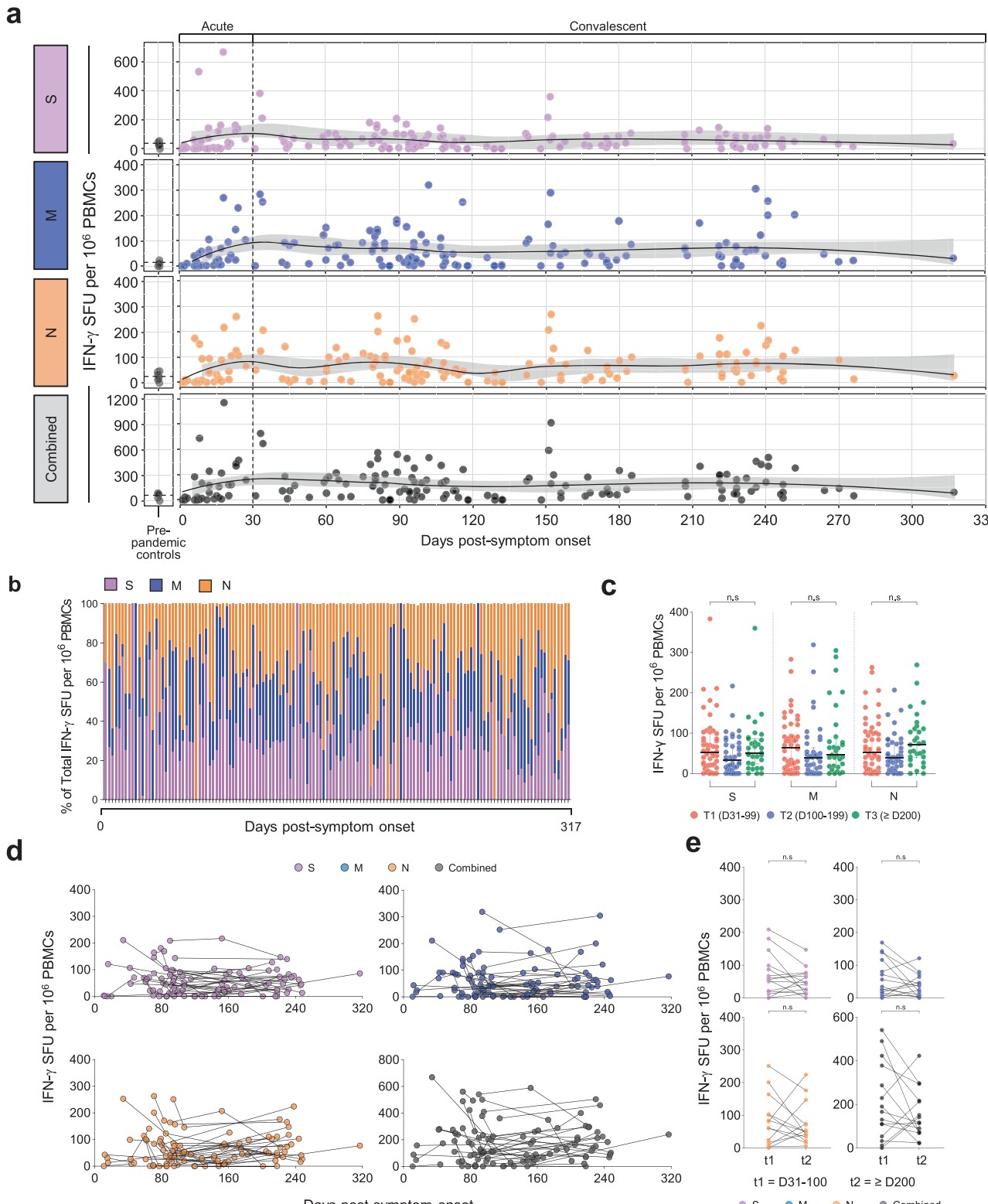

**Fig. 1 SARS-CoV-2-specific IFN-γ responses over 10 months post-infection.** PBMC samples ($n = 159$) from individuals with SARS-CoV-2 infection ($n = 87$) and pre-pandemic PBMC samples ($n = 8$) from healthy donors ($n = 8$) were stimulated with OLPs of S, M, or N (1 μg/mL) for 24 h and the spot-forming units of IFN-γ-secreting cells examined by ELISpot. **a** Scatter plots showing the relationship between DPSO and IFN-γ responses. The black line is a LOESS smooth nonparametric function, and the gray shading represents the 95% confidence interval. **b** The composition of S-, M-, or N-specific IFN-γ responses among the total IFN-γ responses in each individual. **c** IFN-γ responses were compared between T1 ($n = 49$, 31–99 DPSO), T2 ($n = 41$, 100–199 DPSO), and T3 ($n = 31$, ≥200 DPSO). Data are presented as median and interquartile range (IQR). **d, e** IFN-γ responses were analyzed in longitudinally tracked samples ($n = 103$) from 39 individuals. **d** Scatter plots showing the relationship between DPSO and IFN-γ responses. **e** IFN-γ responses were compared between paired samples at two-time points ($n = 15$; t1, 31–100 DPSO; t2, ≥200 DPSO). Statistical analysis was performed using the two-sided Kruskal–Wallis test with Dunns' multiple comparisons test (**c**) or the Wilcoxon signed-rank test (**e**). n.s, not significant.

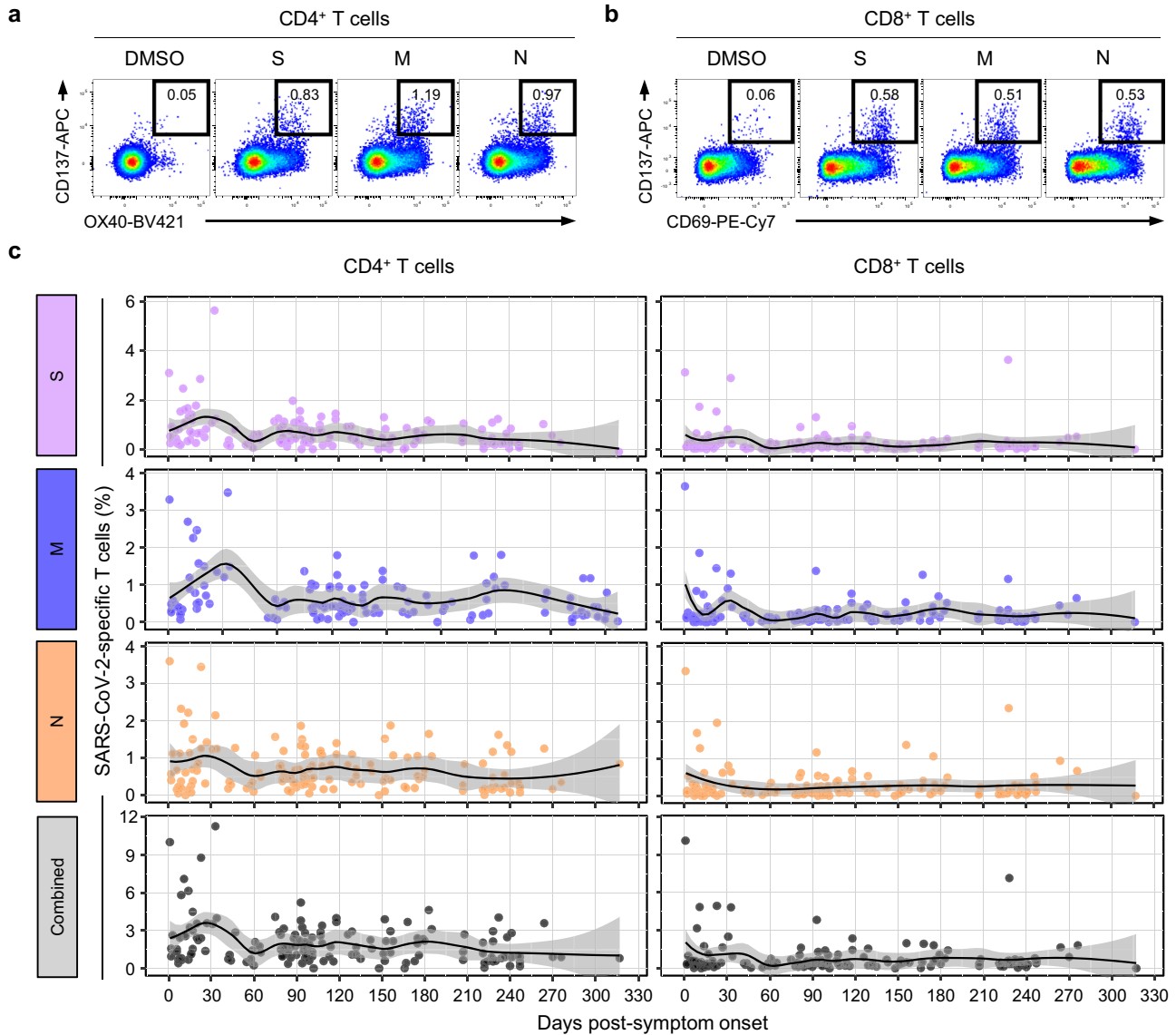

**Fig. 2 Kinetics of SARS-CoV-2-specific activation-induced marker (AIM)$^+$ T cells.** PBMC samples ($n = 146$) from individuals with SARS-CoV-2 infection ($n = 82$) were stimulated with OLPs of S, M, or N (1 µg/mL) for 24 h. The frequency of AIM$^+$ (CD137$^+$OX40$^+$) cells among CD4$^+$ T cells and the frequency of AIM$^+$ (CD137$^+$CD69$^+$) cells among CD8$^+$ T cells were analyzed. Representative flow cytometry plots showing the frequency of AIM$^+$ cells among CD4$^+$ (**a**) or CD8$^+$ (**b**) T cells. **c** Scatter plots showing the relationship between DPSO and the frequency of AIM$^+$ cells among CD4$^+$ (left) or CD8$^+$ (right) T cells. The black is a LOESS smooth nonparametric function, and the gray shading represents the 95% confidence interval.

DPSO), the level of RBD-specific antibodies significantly decreased but the level of neutralizing activity did not (Supplementary Fig. 4b). We did not observe a significant correlation between long-term (≥200 DPSO) SARS-CoV-2-specific T cell responses as evaluated by IFN-γ ELISpot and AIM assays and antibody levels (Supplementary Fig. 4c, d). Collectively, these results indicate that SARS-CoV-2-specific T cell responses are long-lasting over 10 months in COVID-19 convalescent patients although SARS-CoV-2-specific antibody response may decrease.

**SARS-CoV-2-specific T$_{SCM}$ cells develop after the infection.** Next, we examined the differentiation status of SARS-CoV-2-specific AIM$^+$ T cells based on CCR7 and CD45RA expression in CD4+ (Fig. 3a) and CD8$^+$ (Fig. 3b) T cell populations. Among SARS-CoV-2-specific AIM$^+$CD4$^+$ T cells, the proportion of CCR7$^+$CD45RA$^-$ (T$_{CM}$) cells was maintained at approximately 50% on average during the study period, and the proportion of CCR7$^-$CD45RA$^-$ (T$_{EM}$)

cells increased up to ~35% on average until 60 DPSO and then maintained thereafter (Fig. 3c). CCR7$^-$CD45RA$^+$ (T$_{EMRA}$) cells were a minor population (<5%; Fig. 3c). Notably, the proportion of CCR7$^+$CD45RA$^+$ cells, which include both naïve and T$_{SCM}$ cells, was ~30% on average in the acute phase, decreasing to ~10% on average by 60 DPSO and then maintained thereafter (Fig. 3c).

Among SARS-CoV-2-specific AIM$^+$CD8$^+$ T cells, the proportion of CCR7$^+$CD45RA$^-$ (T$_{CM}$) cells was ~20% on average in the acute phase and gradually decreased during the study period (Fig. 3d). The proportion of CCR7$^-$CD45RA$^-$ (T$_{EM}$) cells increased up to 50% on average until 60 DPSO and was maintained thereafter (Fig. 3d). The proportion of CCR7$^-$CD45RA$^+$ (T$_{EMRA}$) cells and CCR7$^+$CD45RA$^+$ cells was maintained (~25% and 25–35% on average, respectively) during the study period (Fig. 3d).

We further investigated whether CCR7$^+$CD45RA$^+$ cells include T$_{SCM}$ cells, which have a self-renewal capacity and multipotency for differentiation into diverse T cell subsets, by examining CD95, a marker of T$_{SCM}$ cells[15,16]. In both AIM$^+$CD4$^+$ and AIM$^+$CD8$^+$

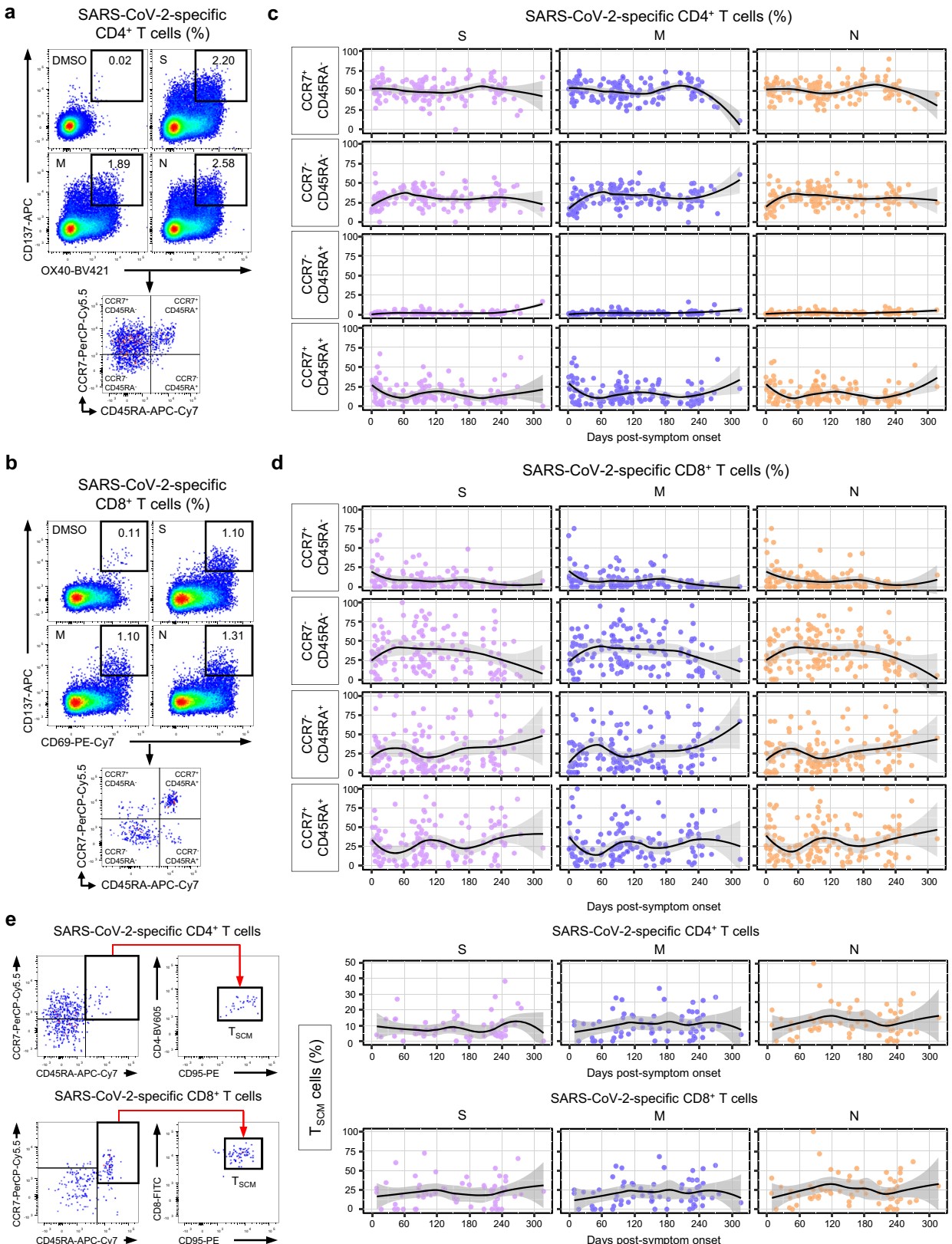

T cells, CD95+ cells were a dominant population among CCR7+ CD45RA+ cells (Fig. 3e), indicating that the CCR7+CD45RA+ cells among AIM+CD4+ or AIM+CD8+ T cells are mainly $T_{SCM}$ cells. The frequency of $T_{SCM}$ cells among both AIM+CD4+ and AIM+CD8+ T cells gradually increased until 120 DPSO, when it reached a stable plateau (Fig. 3e). We found no significant difference in the

frequency of $T_{SCM}$ cells among AIM+CD4+ and AIM+CD8+ T cells between T1 (31–99 DPSO), T2 (100–199 DPSO), and T3 (≥200 DPSO; Supplementary Fig. 5a, b).

**Successful development of SARS-CoV-2-specific $T_{SCM}$ cells is confirmed by direct ex vivo MHC-I multimer staining.** To

**Fig. 3 Differentiation status of SARS-CoV-2-specific AIM$^+$ T cells. a-d** PBMC samples ($n = 146$) from individuals with SARS-CoV-2 infection ($n = 82$) were stimulated with OLPs of S, M, or N (1 μg/mL) for 24 h and the expression of CCR7 and CD45RA was analyzed in AIM$^+$ (CD137$^+$OX40$^+$) CD4$^+$ (**a, c**) and AIM$^+$ (CD137$^+$CD69$^+$) CD8$^+$ (**b, d**) T cells. Gating strategies for identifying each memory subset among AIM$^+$CD4$^+$ (**a**) or AIM$^+$CD8$^+$ (**b**) T cells. Scatter plots showing the relationship between DPSO and the proportion of the indicated subsets among AIM$^+$CD4$^+$ (**c**) or AIM$^+$CD8$^+$ (**d**) T cells. **e** PBMC samples ($n = 68$) from COVID-19 convalescent patients ($n = 59$) were stimulated with OLPs of S, M, or N (1 μg/mL) for 24 h and the frequency of T$_{SCM}$ (CCR7$^+$CD45RA$^+$CD95$^+$) cells was analyzed in AIM$^+$CD4$^+$ (upper) and AIM$^+$CD8$^+$ (lower) T cells. Left, The gating strategy for identifying T$_{SCM}$ cells. Right, Scatter plots showing the relationship between DPSO and the proportion of T$_{SCM}$ cells among AIM$^+$CD4$^+$ or AIM$^+$CD8$^+$ T cells. The black line is a LOESS smooth nonparametric function, and the gray shading represents the 95% confidence interval (**c, d, e**).

validate the results from in vitro stimulation-based AIM assays, we detected SARS-CoV-2-specific CD8$^+$ T cells by performing direct ex vivo MHC-I multimer staining and examined the differentiation status of MHC-I multimer$^+$ cells. We used an HLA-A*02 multimer loaded with SARS-CoV-2 S$_{269}$ (YLQPRTFLL) peptide that has a low degree of homology to the human common cold coronaviruses (ccCoVs)[22,23]. MHC-I multimer$^+$ cells were detected during the study period until 234 DPSO (Fig. 4a, b), and we determined the proportions of T$_{CM}$ (CCR7$^+$CD45RA$^-$), T$_{EM}$ (CCR7$^-$CD45RA$^-$), T$_{EMRA}$ (CCR7$^-$CD45RA$^+$), and T$_{SCM}$ (CCR7$^+$CD45RA$^+$CD95$^+$) cells among MHC-I multimer$^+$ cells (Fig. 4c). Influenza A virus (IAV)- and cytomegalovirus (CMV)-specific MHC-I multimers were also used to stain PBMCs from healthy donors. Similar to the data from AIM$^+$CD8$^+$ T cells, T$_{EM}$ and T$_{EMRA}$ cells were the dominant populations among SARS-CoV-2-specific MHC-I multimer$^+$ cells, whereas T$_{CM}$ cells were a minor population during the study period (Fig. 4d). Among IAV- and CMV-specific MHC-I multimer$^+$ cells from healthy donors, T$_{EM}$ and T$_{EMRA}$ cells were dominantly present, whereas T$_{CM}$ cells were scarcely detected. T$_{SCM}$ cells were also detected among SARS-CoV-2-specific MHC-I multimer$^+$ cells during the study period, but not among IAV- and CMV-specific cells (Fig. 4d). In particular, the frequency of T$_{SCM}$ cells among SARS-CoV-2-specific MHC-I multimer$^+$ cells was high 60–125 DPSO.

A recent study has revealed two distinct subsets of CCR7$^+$ stem cell-like progenitors: CCR7$^+$PD-1$^-$TIGIT$^-$ cells with stem cell-like features and CCR7$^+$PD-1$^+$TIGIT$^+$ cells with exhausted traits[23,24]. Therefore, we compared the percentage of PD-1$^-$TIGIT$^-$ cells among MHC-I multimer$^+$CD8$^+$ T$_{SCM}$ cells to the percentage among MHC-I multimer$^+$CD8$^+$ T cells. We found that the frequency of PD-1$^-$TIGIT$^-$ cells was significantly higher among MHC-I multimer$^+$CD8$^+$ T$_{SCM}$ cells (Fig. 4e), confirming that SARS-CoV-2-specific T$_{SCM}$ cells are bona fide stem-like memory cells.

We performed additional experiments to examine the detailed stem cell-like properties of SARS-CoV-2-specific T$_{SCM}$ cells. We used PBMCs obtained from 110 to 140 DPSO ($n = 3$) and sorted SARS-CoV-2-specific AIM$^+$(CD137$^+$) CD8$^+$ T$_{SCM}$ (CCR7$^+$CD45RA$^+$CD95$^+$) cells. Next, we labeled them with CellTrace Violet (CTV), stimulated them with the S OLP pool for 7 days, and assessed their proliferation. We observed robust proliferation of SARS-CoV-2-specific T$_{SCM}$ cells (Supplementary Fig. 6a). We also examined whether SARS-CoV-2-specific T$_{SCM}$ cells differentiate into other memory subsets following stimulation with the S OLP pool for 7 days and found that T$_{SCM}$ cells differentiated into diverse memory subsets, comprising ~72% T$_{EM}$, 10% T$_{EMRA}$, 7% T$_{CM}$, and 10% T$_{SCM}$ cells (Supplementary Fig. 6b). We also evaluated the self-renewal capacity of T$_{SCM}$ cells following stimulation with IL-15, a homeostatic cytokine, for 5 days and observed robust proliferation of SARS-CoV-2-specific CD4$^+$ and CD8$^+$ T$_{SCM}$ cells (Supplementary Fig. 6c). Based on our results that SARS-CoV-2-specific T$_{SCM}$ cells exhibit a potent proliferative capacity and multipotency to reconstitute diverse memory subsets, we anticipate that SARS-CoV-2-specific memory T cells can robustly proliferate and differentiate into effector cells upon re-exposure to SARS-CoV-2.

**Polyfunctionality and proliferation capacity are preserved in long-term SARS-CoV-2-specific T cells.** As we observed the generation of SARS-CoV-2-specific T$_{SCM}$ cells with self-renewal capacity and multipotency, we aimed to examine the kinetics of the polyfunctionality of SARS-CoV-2-specific T cells during the study period. To this end, we performed intracellular cytokine staining of IFN-γ, IL-2, TNF, and CD107a following stimulation with OLP pools of S, M, and N and analyzed the polyfunctionality of CD4$^+$ and CD8$^+$ T cells (Fig. 5a). We defined polyfunctional cells as cells exhibiting positivity for ≥2 effector functions. Among SARS-CoV-2-specific CD4$^+$ and CD8$^+$ T cells, the average proportion of polyfunctional cells was 25–40% and 30–50%, respectively, 60 DPSO, and was maintained until 254 DPSO (Fig. 5b). We found no significant difference in the frequency of polyfunctional cells among SARS-CoV-2-specific CD4$^+$ and CD8$^+$ T cells between T1 (31–99 DPSO), T2 (100–199 DPSO), and T3 (≥200 DPSO; Fig. 5c). Preserved polyfunctionality among SARS-CoV-2-specific CD4$^+$ and CD8$^+$ T cells was also observed when polyfunctionality was evaluated according to the number of positive effector functions (Fig. 5d).

We also examined the antigen-induced proliferation capacity of long-term SARS-CoV-2-specific memory T cells. We performed CTV dilution assays and Ki-67 staining using PBMCs obtained after 200 DPSO. CD4$^+$ and CD8$^+$ T cells exhibited a significant proliferative response following in vitro stimulation with the S OLP pool (Fig. 5e), indicating that SARS-CoV-2-specific memory T cells elicit rapid recall responses upon viral re-exposure.

Considering preserved polyfunctionality and proliferation capacity of SARS-CoV-2-specific memory T cells, the current data indicate that memory T cells contribute to protective immunity against re-infection even 10 months after the primary infection.

**SARS-CoV-2-specific T cell memory is maintained regardless of disease severity.** To determine the effect of disease severity at the time of acute infection on the generation of long-term (≥200 DPSO) SARS-CoV-2-specific T cell memory, we compared long-term SARS-CoV-2-specific T cell responses between the asymptomatic/mild patient group and the moderate/severe/critical patient group. In this comparison, we analyzed the magnitude of the response as evaluated by IFN-γ ELISpot and AIM assays, and the proportion of polyfunctional cells and T$_{SCM}$ cells among SARS-CoV-2-specific memory T cells. We found no significant difference in any of the parameters between the two groups (Supplementary Fig. 7a–d).

A recent study showed that 6 months after infection, asymptomatic individuals have lower SARS-CoV-2-specific T cell responses than individuals who recovered from symptomatic infection[25]. Another study reported similar SARS-CoV-2-specific T cell responses between asymptomatic individuals and symptomatic COVID-19 patients, but that the T cell responses may decline more rapidly in asymptomatic individuals[26]. Therefore, we compared long-term (≥200 DPSO) T cell memory between the asymptomatic and symptomatic groups. We could not find a

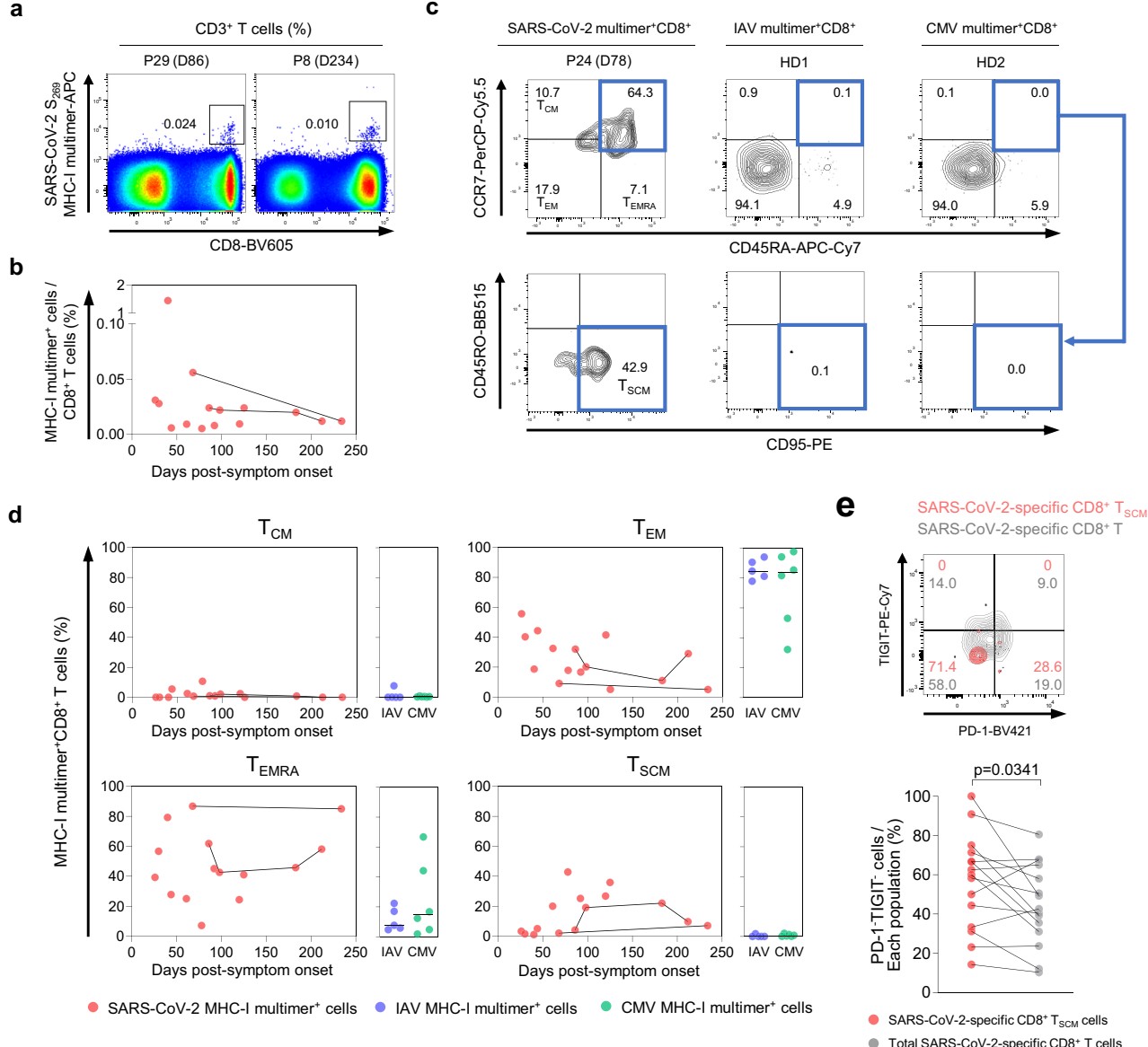

**Fig. 4 Frequency and differentiation status of SARS-CoV-2-specific MHC-I multimer⁺ T cells.** PBMC samples ($n = 15$) from individuals with SARS-CoV-2 infection ($n = 11$) were analyzed by flow cytometry. **a** Representative flow cytometry plots showing the ex vivo detection of SARS-CoV-2 $S_{269}$ multimer⁺CD8⁺ T cells in the gate of CD3⁺ T cells. **b** Scatter plot showing the relationship between DPSO and the frequency of SARS-CoV-2 $S_{269}$ multimer⁺ cells among total CD8⁺ T cells. Samples from the same patient are connected by solid lines. The expression of CCR7, CD45RA, and CD95 was analyzed in SARS-CoV-2 $S_{269}$ multimer⁺CD8⁺ T cells. IAV $MP_{58}$ multimer⁺ ($n = 5$) and CMV $pp65_{495}$ multimer⁺ ($n = 6$) cells from the PBMCs of healthy donors were also analyzed. Representative flow cytometry plots (**c**) show the proportion of the indicated subsets among multimer⁺ cells, and scatter plots (**d**) show the relationship between DPSO and the proportion of the indicated subsets among SARS-CoV-2 $S_{269}$ multimer⁺ cells. Samples from the same patient are connected by solid lines. Summary data showing the proportion of the indicated subsets among IAV multimer⁺ and CMV multimer⁺ cells are also presented (**d**). Horizontal lines represent median. **e** A representative flow cytometry plot (upper) and summary data (lower) showing the percentage of PD-1⁻TIGIT⁻ cells among SARS-CoV-2 $S_{269}$ multimer⁺CD8⁺ $T_{SCM}$ cells and total SARS-CoV-2 $S_{269}$ multimer⁺CD8⁺ T cells. Statistical analysis was performed using the two-sided Wilcoxon signed-rank test (**e**).

significant difference in the number of IFN-γ spots between the two groups (Supplementary Fig. 8). In addition, when we performed subgroup analyses of the asymptomatic/mild group and moderate/severe/critical group, we found no significant differences in the number of IFN-γ spots or the frequencies of AIM⁺ cells among T1 (31–99 DPSO), T2 (100–199 DPSO), and T3 (≥200 DPSO) in either group (Supplementary Fig. 9). From these results, we conclude that SARS-CoV-2-specific T cell memory is successfully maintained regardless of the severity of COVID-19.

## Discussion

Memory T cells play a crucial role in viral clearance during re-infection, but the longevity and differentiation status of SARS-CoV-2-specific memory T cells among COVID-19 convalescent patients remain unclear. In the present study, we demonstrated that SARS-CoV-2-specific memory T cell responses were maintained in COVID-19 convalescent patients 10 months post-infection regardless of the disease severity. Notably, we found that SARS-CoV-2-specific $T_{SCM}$ cells were successfully developed, indicating that SARS-CoV-2-specific T cell memory may be long-

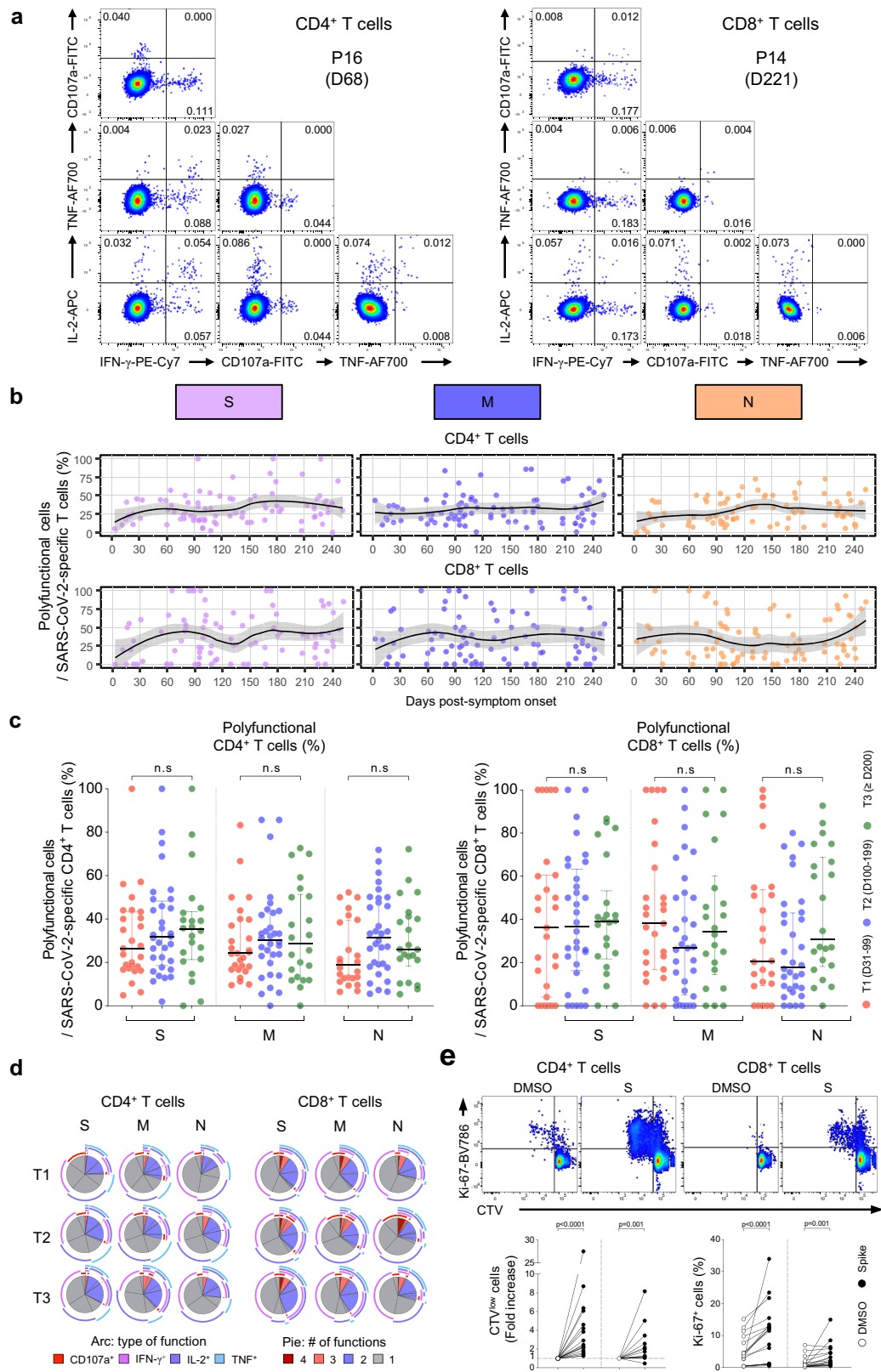

lasting in COVID-19 convalescent patients. These findings were supported by the SARS-CoV-2-specific T cells from PBMCs obtained after 200 DPSO exhibiting sustained polyfunctionality and proliferation capacity. Our results may fill a gap in understanding T cell memory responses after recovery from SARS-CoV-2 infection.

Recent studies suggest critical roles of T cells in the clearance of SARS-CoV-2 and protection from developing severe COVID-19. In particular, it has been shown that coordination of adaptive immune responses, including CD4$^+$ T cell, CD8$^+$ T cell, and antibody responses, is essential for controlling COVID-19[20]. Notably, peak disease severity has been shown to inversely correlate with the

**Fig. 5 Polyfunctionality and proliferation capacity of SARS-CoV-2-specific T cells.** PBMC samples ($n = 90$) from individuals with SARS-CoV-2 infection ($n = 39$) were stimulated with OLPs of S, M, or N (1 µg/mL) for 6 h. Intracellular cytokine staining was performed to examine the frequency of polyfunctional cells exhibiting positivity for ≥2 effector functions among SARS-CoV-2-specific CD4[+] and CD8[+] T cells. **a** Representative flow cytometry plots showing the frequency of polyfunctional cells among CD4[+] (left) and CD8[+] (right) T cells. **b** Scatter plots showing the relationship between DPSO and the frequency of polyfunctional cells among SARS-CoV-2-specific CD4[+] (upper) or CD8[+] (lower) T cells. The black line is a LOESS smooth nonparametric function, and the gray shading represents the 95% confidence interval. **c** The fraction of polyfunctional cells among SARS-CoV-2-specific CD4[+] (left) or CD8[+] (right) T cells was compared between T1 ($n = 17$, 31–99 DPSO), T2 ($n = 39$, 100–199 DPSO), and T3 ($n = 25$, ≥200 DPSO). Data are presented as median and IQR. **d** Pie charts showing the fraction of cells positive for a given number of functions among SARS-CoV-2-specific CD4[+] (left) or CD8[+] (right) T cells. Each arc in the pie chart represents the indicated function. **e** CTV-labeled PBMCs ($n = 18$) obtained after 200 DPSO were stimulated with S OLP pool (1 µg/mL) for 120 h and the frequency of CTV[low] and Ki-67[+] cells among CD4[+] and CD8[+] T cells was analyzed. Representative plots (upper) and summary data (lower) are presented. Statistical analysis was performed using the two-sided Kruskal–Wallis test with Dunns' multiple comparisons test (**c**) or the two-sided Wilcoxon signed-rank test (**e**). n.s, not significant.

frequency of SARS-CoV-2-specific CD4[+] and CD8[+] T cells instead of SARS-CoV-2 antibody titers[20]. In another study, T cell responses were detected among COVID-19 convalescent patients without detectable SARS-CoV-2 IgG[27]. In a rhesus macaque model, CD8-depleted convalescent animals exhibited limited viral clearance in the respiratory tract upon SARS-CoV-2 re-challenge, suggesting that CD8[+] T cells contribute to viral clearance during SARS-CoV-2 re-infection[10]. Collectively, these studies strongly demonstrate a protective role of T cells in COVID-19.

Currently, we do not have exact information on how long adaptive immune memory lasts in COVID-19 convalescent patients. Whether antibody responses to SARS-CoV-2 wane over time in patients who have recovered from COVID-19 remains controversial[28–30]. Previous studies on SARS-CoV-1 and MERS-CoV infection have shown that T cell responses were more enduring compared to antibody responses[12,13]. A recent study detected SARS-CoV-1-specific T cell responses 17 years after infection[6]. Similarly, slow decay of SARS-CoV-2-specific memory T cells is expected. In the present study, we demonstrated the maintenance of SARS-CoV-2-specific memory T cell responses in COVID-19 convalescent patients over 10 months post-infection using a battery of T cell assays. Given that vaccination programs for prophylaxis of SARS-CoV-2 infection are being launched worldwide, another question is how long memory T cells elicited by vaccination will last. Vaccine-induced memory T cells may differ in phenotype and durability from infection-induced memory T cells, which should be addressed in future studies.

Although this current study showed the persistence of SARS-CoV-2-specific T cell responses in COVID-19 convalescent patients, another recent study suggested a reduction in SARS-CoV-2-specific CD4[+] and CD8[+] T cell responses with a half-life of 3–5 months[14]. This discrepancy can be explained by multiple factors. First, the demographic characteristics of the study cohorts were different. Asians comprised 98% of our study population but only 7% in the other study. SARS-CoV-2-specific T cell responses may vary depending on ethnicity. A demographic difference may also explain why T cell memory is maintained in the current study regardless of disease severity but declined more rapidly in asymptomatic individuals in a previous study[26]. Second, geographic factors also need to be considered, particularly in relation to previous exposure to other coronaviruses. Le Bert et al.[6] demonstrated that SARS-CoV-2-reactive T cell responses are present in 50% of unexposed individuals in Singapore. Interestingly, they showed that the T cell epitopes found among unexposed donors were conserved among animal beta-coronaviruses rather than common-cold coronaviruses. Third, the clinical characteristics of the patients, including peak disease severity, were different between the two studies. Finally, the other study relied on a single type of T cell assay, whereas our study examined T cell memory using multiple types of assays, including IFN-γ ELISpot, AIM, ICS, and proliferation assays. More recent studies support our results by

demonstrating the persistence of SARS-CoV-2-specific T cell responses in COVID-19 convalescent patients[31,32].

A series of studies have reported the existence of SARS-CoV-2-reactive T cell responses among unexposed individuals, suggesting that memory T cells induced by previous ccCoV infection are cross-reactive to SARS-CoV-2 proteins[5,33–36]. Therefore, in the present study, we could not distinguish de novo primed T cells by SARS-CoV-2 infection from pre-existing, cross-reactive ccCoV-specific T cells. However, we could selectively examine de novo primed SARS-CoV-2-specific T cells by staining with an HLA-A*02 multimer loaded with SARS-CoV-2 S269 (YLQPRTFLL) peptide. This epitope peptide has a low degree of homology with ccCoVs, including OC43, HKU1, 229E, and NL63[22,23]. In future studies of SARS-CoV-2-specific T cell responses, cross-reactive epitope peptides and SARS-CoV-2-specific epitope peptides will need to be used separately[34].

Among distinct subsets of memory T cells, T$_{SCM}$ cells possess a superior ability for self-renewal, memory recall responses, and multipotency to reconstitute diverse memory subsets[15]. Therefore, long-term T cell memory relies on the successful generation of T$_{SCM}$ cells[16]. Previous studies showed that long-lasting YFV-specific CD8[+] T cells resemble T$_{SCM}$ cells[17,18]. We and others have also suggested the generation of SARS-CoV-2-specific T$_{SCM}$ cells in the convalescent phase of COVID-19 on the basis of the expression of CCR7 and CD45RA[8,23]. However, these studies on COVID-19 patients did not examine the definitive marker of T$_{SCM}$ cells, CD95. In the present study, we delineated the kinetics of T$_{SCM}$ cells using both AIM assays and MHC-I multimer staining and observed the successful generation of T$_{SCM}$ cells in COVID-19 convalescent patients. In line with these findings, we also found sustained polyfunctionality and proliferation capacity, suggesting efficient memory recall responses. A recent study has proposed that CCR7[+] stem-like progenitor cells are composed of two separate populations, which are distinguished by PD-1 and TIGIT expression[24]. We demonstrated that PD-1 and TIGIT are rarely expressed in SARS-CoV-2-specific T$_{SCM}$ cells, indicating that SARS-CoV-2-specific T$_{SCM}$ cells are not exhausted-like progenitors, but bona fide stem-like memory cells.

Polyfunctional T cells, which exert multiple effector functions simultaneously, play a critical role in host protection against viral infection[37–39]. For example, polyfunctional CD8[+] T cells are preserved in human immunodeficiency virus-infected long-term non-progressors[38]. In addition, polyfunctional T cell responses are associated with effective control of hepatitis C virus (HCV)[39]. It has also been reported that virus-specific polyfunctional T cells can be successfully developed by immunization with vaccinia virus[40] and an HCV vaccine[41]. Collectively, polyfunctional memory T cells control viral infection more efficiently than monofunctional T cells. Therefore, generation of polyfunctional memory T cells following natural infection or vaccination is expected to confer protective immunity. In this regard, the

sustained polyfunctionality of long-term SARS-CoV-2-specific T cells observed in our study is highly suggestive of long-lasting protective immunity in COVID-19 convalescent patients.

In the current study, we conducted a comprehensive analysis of SARS-CoV-2-specific memory T cell responses over 10 months post-infection. Considering that the current study was based on random sample collection, the conclusion needs to be validated by a study based on a systematic blood collection protocol. Despite this limitation, our current analysis provides valuable information regarding the longevity and differentiation of SARS-CoV-2-specific memory T cells elicited by natural infection. These data add to our basic understanding of memory T cell responses in COVID-19, which aids in establishing an effective vaccination program and epidemiological measurement.

## Methods

**Patients and specimens**. In this study, 101 patients with PCR-confirmed SARS-CoV-2 infection were enrolled from Ansan Hospital and Chungbuk National University Hospital, Republic of Korea. Peripheral blood was obtained from all patients with SARS-CoV-2 infection. In seven asymptomatic patients, the date of the first admission was regarded as seven DPSO because the date of the first admission among symptomatic patients was an average seven DPSO. We also used PBMCs obtained before the COVID-19 pandemic from eight healthy donors. This study was reviewed and approved by the institutional review board of all participating institutions (Chungbuk National University Hospital: 2020-03-036-001; Korea University Ansan Hospital: 2020AS0122) and conducted according to the principles of the Declaration of Helsinki. Informed consent was obtained from all donors and patients.

PBMCs were isolated by density gradient centrifugation using Lymphocyte Separation Medium (Corning). After isolation, the cells were cryopreserved in fetal bovine serum (FBS; Corning) with 10% dimethyl sulfoxide (DMSO; Sigma-Aldrich) until use.

**Ex vivo IFN-γ enzyme-linked immunospot assay**. Plates with hydrophobic polyvinylidene difluoride membrane (Millipore) were coated with 2 μg/mL anti-human monoclonal IFN-γ coating antibody (clone 1-D1K, Mabtech) overnight at 4 °C. The plates were washed with sterile phosphate-buffered saline (PBS) and blocked with 1% bovine serum albumin (Bovogen) for 1 h at room temperature (RT). 700,000 PBMCs were seeded per well and stimulated with 1 μg/mL OLP pools spanning SARS-CoV-2 S, M, and N proteins (Miltenyi Biotec) for 24 h at 37 °C. We used 10 μg/mL phytohemagglutinin as a positive control and an equimolar amount of DMSO as a negative control. Plates were washed with 0.05% Tween-PBS (Junsei Chemical) and incubated with 0.25 μg/mL biotinylated anti-human monoclonal IFN-γ antibody (clone 7-B6-1, Mabtech) for 2 h at RT. After washing, streptavidin-alkaline phosphatase (Invitrogen) was added sequentially. Precipitates were detected with AP color reagent (Bio-Rad) and the reaction stopped by rinsing with distilled water. Spot-forming units were quantified using an automated ELI-Spot reader (AID). To quantify SARS-CoV-2-specific responses, spots in the negative control wells were subtracted from the OLP-stimulated wells.

**Multi-color flow cytometry**. Cells were stained with fluorochrome-conjugated antibodies for specific surface markers for 10 min at RT. Dead cells were excluded using LIVE/DEAD red fluorescent reactive dye (Invitrogen). In intracellular staining experiments, cells were fixed and permeabilized using the FoxP3 staining buffer kit (Invitrogen), and then stained for intracellular markers for 30 min at 4 °C. The following monoclonal antibodies were used for multi-color flow cytometry: anti-hCD3 BV510 (clone UCHT1, cat# 563109, 1:100), anti-hCD3 BV786 (clone UCHT1, cat# 565491, 1:100), anti-hCD4 (clone RPA-T4, cat# 562358, 1:100), anti-hCD4 FITC (clone RPA-T4, cat# 555346, 1:100), anti-hCD4 PerCP™Cy5.5 (clone RPA-T4, cat# 560650, 1:100), anti-hCD8 APC-Cy7 (clone SK1, cat# 560179, 1:100), anti-hCD8 BV605 (clone SK1, cat# 564116, 1:100), anti-hCD8 BV711 (clone RPA-T8, cat# 563677, 1:100), anti-hCD14 PE-CF594 (clone MφP9, cat# 562335, 1:100), anti-hCD19 PE-CF594 (clone HIB19, cat# 562294, 1:100), anti-hCD27 BV510 (clone L128, cat# 563092, 1:100), anti-hCD45RO BB515 (clone UCHL1, cat# 564529, 1:100), anti-hCD69 PE-Cy7 (clone FN50, cat# 557745, 1:100), anti-hCD95 PE (clone DX2, cat# 555674, 1:100), anti-hCD107a FITC (clone H4A3, cat# 555800, 1:100), anti-hCD137 APC (clone 4B4-1, cat# 550890, 1:100), anti-hCD137 BV421 (clone 4B4-1, cat# 564091, 1:100), anti-hCD154 APC (clone TRAP1, cat# 555702, 1:100), anti-hIFN-γ PE-Cy7 (clone 4S.B3, cat# 557844, 1:100), anti-hIL-2 APC (clone MQ1-17H12, cat# 554567, 1:100), anti-hKi-67 BV786 (clone B56, cat# 563756, 1:100), and anti-hTNF AF700 (clone Mab11, cat# 557996, 1:100) from BD Biosciences; anti-hCCR7 PerCP™Cy5.5 (clone G043H7, cat# 353220, 1:100), anti-hCD3 APC (clone HIT3a, cat# 300312, 1:100), anti-hCD8 FITC (clone RPA-T8, cat# 301050, 1:100), anti-hCD45RA APC-Cy7 (clone HI100, cat# 304128, 1:100), anti-hCD137 PE (clone 4B4-1, cat# 309804, 1:100), anti-hOX40 BV421 (clone Ber-ACT35, cat# 350014, 1:100), and anti-hPD-1 BV421 (clone EH12.2H7, cat# 329920, 1:100) from BioLegend; anti-hTIGIT PE-Cy7 (clone MBSA43, cat# 25-

9500-42, 1:100) from Invitrogen. The following multimers were used for multi-color flow cytometry: GILGFVFTL (IAV MP58) HLA-A*0201 APC Dextramer (cat# WB2161, 1:20) and NLVPMVATV (CMV pp65495) HLA-A*0201 APC Dextramer (cat# WB2132, 1:20) from Immudex; YLQPRTFLL (SARS-CoV-2 S269) HLA-A*0201 APC Pentamer (cat# 4339, 1:20) from Proimmune. Multi-color flow cytometry was performed using an LSR II instrument with FACSDiva (BD Biosciences) and the data analyzed in FlowJo software (FlowJo LLC). The details of fluorochrome-conjugated MHC-I multimers and monoclonal antibodies used in this study are described in Supplementary Table 3.

**Activation-induced marker assay**. PBMCs were blocked with 0.5 μg/mL anti-human CD40 mAb (clone HB14, Miltenyi Biotec) in RPMI 1640 supplemented with 10% FBS and 1% penicillin and streptomycin (Welgene) for 15 min at 37 °C. The cells were then cultured in the presence of 1 μg/mL SARS-CoV-2 OLP pools and 1 μg/mL anti-human CD28 and CD49d mAbs (clone L293 and L25, respectively, BD Biosciences) for 24 h. Stimulation with an equal concentration of DMSO in PBS was performed as a negative control.

**MHC-I multimer staining**. PBMCs were stained with multimers for 15 min at RT and washed twice. Additional surface markers were stained using the protocols described above.

**Stimulation for intracellular cytokine staining**. PBMCs were cultured in the presence of SARS-SoV-2 OLP pools and 1 μg/mL anti-human CD28 and CD49d mAbs for 6 h at 37 °C. Brefeldin A (GolgiPlug, BD Biosciences) and monensin (GolgiStop, BD Biosciences) were added 1 h after the initial stimulation. Poly-functionality of the stimulated cells was analyzed with SPICE software.

**Proliferation assays**. PBMCs were labeled using the CellTrace™ Violet Cell Proliferation Kit (Invitrogen) at a concentration of $1.0 \times 10^6$ cells/mL for 20 min at 37 °C. We then added 1% FBS-PBS to the cells for 5 min at RT to quench unbound dyes. Cells were washed and cultured in RPMI 1640 containing 10% FBS and 1% penicillin-streptomycin at a concentration of 500,000 cells per well in the presence of 1 μg/mL SARS-CoV-2 S peptide pools and 1 μg/mL anti-human CD28 and CD49d mAbs for 120 h. Stimulation with an equal concentration of DMSO in PBS was performed as a negative control. After incubation, cells were harvested and stained with antibodies for analysis by flow cytometry.

**T_SCM cell sorting and functional analysis**. AIM assays were performed with SARS-CoV-2 S OLP stimulation, and CD8+ T cells were magnetically enriched using CD8 MicroBeads (Miltenyi Biotec). AIM+(CD137+) CD8+ T_SCM(CCR7+ CD45RA+CD95+) cells were further sorted using FACS ARIA II cell sorter (BD Biosciences) with purity of >95%. Sorted CD8+ T_SCM cells were labeled with CTV and cultured with 1 μg/mL SARS-CoV-2 S OLPs in the presence of 1 μg/mL anti-human CD28 and CD49d mAbs for 7 days. Irradiated autologous CD8− cells were used as feeder cells. Proliferation of T_SCM cells and memory phenotypes of the T_SCM cell progeny were assessed by flow cytometry.

For evaluation of the self-renewal capacity of T_SCM cells, CTV-labeled PBMCs were cultured in the presence of recombinant human IL-15 (25 ng/mL; PeproTech) for 120 h. After incubation, AIM assays were performed with SARS-CoV-2 S OLP stimulation for the detection of SARS-CoV-2 S-specific T_SCM cells. Proliferation of SARS-CoV-2 S-specific CD4+ and CD8+ T_SCM cells was assessed by flow cytometry.

**Enzyme-linked immunosorbent assay**. For the measurement of anti-SARS-CoV-2 RBD IgG levels in plasma samples, SARS-CoV-2-Spike S1-RBD IgG and IgM ELISA Detection Kit (GenScript) was used according to the manufacturer's instruction. For the measurement of SARS-CoV-2 neutralizing activities of plasma samples, SARS-CoV-2 Surrogate Virus Neutralization Detection Kit (GenScript) was used according to the manufacturer's instruction. The percent of the neutralizing activity was calculated with the following equation.

$$\text{Neutralization}(\%) = \left(1 - \frac{\text{Sample absorbance}}{\text{Negative control absorbance}}\right) \times 100\%$$

**Data quantification and statistical analysis**. Data were calculated as background-subtracted data. Background-subtracted data were derived by subtracting the value after DMSO stimulation. When three stimuli were combined, the value after each stimulation was combined and we subtracted triple the value derived from DMSO stimulation.

Statistical analyses were performed using GraphPad Prism version 9 for Windows (GraphPad Software). Significance was set at $p < 0.05$. The Wilcoxon signed-rank test was used to compare data between two paired groups. The Mann–Whitney U test was used in comparison between two unpaired groups. In addition, the Kruskal-Wallis test with Dunns' multiple comparisons test was used to compare non-parametric data between multiple unpaired groups. To assess the significance of correlation, the Spearman correlation test was used. In cross-sectional analyses, a non-parametric local regression (LOESS) function was

employed with 95% confidence interval. LOESS analysis was performed using R[42] with following packages: dplyr, tidyr, ggplot2, tidyverse, ggthemes, and svglite.

**Reporting summary**. Further information on research design is available in the Nature Research Reporting Summary linked to this article.

## Data availability

All data supporting the findings of this study are available within the main manuscript and the supplementary files or provided upon reasonable request. Raw data corresponding to all the main and supplementary figures are included in the Source Data file. Source data are provided with this paper.

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

## Acknowledgements

This work was supported by the Samsung Science and Technology Foundation under Project Number SSTF-BA1402-51, and the Mobile Clinic Module Project funded by KAIST.

## Author contributions

J.H.J., M.-S.R., H.W.J., W.S.C., and E.-C.S. designed the research. H.K.C., J.H.J., H.S., D. W.P., H.W.J., and W.S.C. collected clinical specimens. J.H.J., M.-S.R., and M.S. performed experiments. J.H.J., M.-S.R., S.-H.P., and E.-C.S. analyzed the results. J.H.J., M.-S. R. and E.-C.S. wrote the manuscript.

## Competing interests

The authors declare no competing interests.
