## [Peer Review File · Nature Communications]

REVIEWERS' COMMENTS

Reviewer #1 (Remarks to the Author):

Authors have provided detailed rebuttal to my comments and have addressed most of the major issues raised in my original review. One of my major criticism was that the follow up studies were based on random sample collection rather than a systematic blood collection protocol. It is clear authors are unable to address this issue as it will be impossible to go back start new study. It will important to discuss this limitation of the study in the discussion section and recommend appropriate steps to overcome these limitations. This will help other researchers to plan their study correctly.

I am little surprised that authors did not see any difference in the T cell response from mild/asymptomatic vs severe COVID patients. This may be reflection of the demographic profile of patients.

Overall, I am satisfied that authors have addressed all major issues but the limitations of the data can be discussed in the text.

Reviewer #2 (Remarks to the Author):

The authors have addressed my comments. In particular they now analyzed the results of T cell frequency in relation to the clinical severity of COVID-19 . They show that persistence of memory SARS-CoV-2 specific T cells is similar between patients with different disease severity. However the authors wrote in the results section that " (lines 239-241) recent studies reported that asymptomatic individuals have lower SARS-CoV-2-specific T-cell responses compared to individuals who recovered from symptomatic infection^{25,26} ".

This is not correct, these studies suggest that memory T cells might decline more in asymptomatic infection , but not that asymptomatic individuals have lower SARS-CoV-2 T cell response than symptomatic ones . At the opposite , in particular , reference 26 show that such response were similar or even more robust in asymptomatic. This comment must be changed. What is novel in the data presented is the fact that also over time the frequency of T cells is remarkably similar between patients with different clinical severity.

Reviewer #3 (Remarks to the Author):

The authors have addressed most of my comments.
Just one minor issue, in supplementary Fig 6b, the label should be Tem/Tcm/Temra/Tn.